# Coherent Gradients: An Approach to Understanding Generalization in Gradient Descent-based Optimization

**Satrajit Chatterjee**
Google AI
Mountain View, CA 94043, USA
schatter@google.com

## Abstract

An open question in the Deep Learning community is why neural networks trained with Gradient Descent generalize well on real datasets even though they are capable of fitting random data. We propose an approach to answering this question based on a hypothesis about the dynamics of gradient descent that we call *Coherent Gradients*: Gradients from similar examples are similar and so the overall gradient is stronger in certain directions where these reinforce each other. Thus changes to the network parameters during training are biased towards those that (locally) simultaneously benefit many examples when such similarity exists. We support this hypothesis with heuristic arguments and perturbative experiments and outline how this can explain several common empirical observations about Deep Learning. Furthermore, our analysis is not just descriptive, but prescriptive. It suggests a natural modification to gradient descent that can greatly reduce overfitting.

## 1 Introduction and Overview

Neural networks used in practice often have sufficient effective capacity to learn arbitrary maps from their inputs to their outputs. This is typically demonstrated by training a classification network that achieves good test accuracy on a real dataset $S$, on a modified version of $S$ (call it $S'$) where the labels are randomized and observing that the training accuracy on $S'$ is very high, though, of course, the test accuracy is no better than chance (Zhang et al., 2017). This leads to an important open question in the Deep Learning community (Zhang et al. (2017); Arpit et al. (2017); Bartlett et al. (2017); Kawaguchi et al. (2017); Neyshabur et al. (2018); Arora et al. (2018); Belkin et al. (2019); Rahaman et al. (2019); Nagarajan & Kolter (2019), etc.): Among all maps that fit a real dataset, how does Gradient Descent (GD) find one that generalizes well? This is the question we address in this paper.

We start by observing that this phenomenon is not limited to neural networks trained with GD but also applies to Random Forests and Decision Trees. However, there is no mystery with trees: A typical tree construction algorithm splits the training set recursively into similar subsets based on input features. If no similarity is found, eventually, each example is put into its own leaf to achieve good training accuracy (but, of course, at the cost of poor generalization). Thus, trees that achieve good accuracy on a randomized dataset are much larger than those on a real dataset (e.g. Chatterjee & Mishchenko (2019, Expt. 5)).

Is it possible that something similar happens with GD? We believe so. The type of randomized-label experiments described above show that if there are common patterns to be found, then GD finds them. If not, it fits each example on a case-by-case basis. The question then is, what is it about the dynamics of GD that makes it possible to extract common patterns from the data? And what does it mean for a pattern to be common?

Since the only change to the network parameters in GD comes from the gradients, the mechanism to detect commonality amongst examples must be through the gradients. We propose that this commonality detection can be explained as follows:

1. Gradients are *coherent*, i.e, similar examples (or parts of examples) have similar gradients (or similar components of gradients) and dissimilar examples have dissimilar gradients.

2. Since the overall gradient is the sum of the per-example gradients, it is stronger in directions where the per-example gradients are similar and reinforce each other and weaker in other directions where they are different and do not add up.

3. Since network parameters are updated proportionally to gradients, they change faster in the direction of stronger gradients.

4. Thus the changes to the network during training are biased towards those that simultaneously benefit many examples instead of a few (or one example).

For convenience, we refer to this as the Coherent Gradients hypothesis.

It is instructive to work through the proposed mechanism in the context of a simple thought experiment. Consider a training set with two examples $a$ and $b$. At some point in training, suppose the gradient of $a$, $g_a$, can be decomposed into two orthogonal components $g_{a_1}$ and $g_{a_2}$ of roughly equal magnitude, i.e., there are two, equally good, independent ways in which the network can better fit $a$ (by using say two disjoint parts of the network). Likewise, for $b$. Now, further suppose that one of the two ways is common to both $a$ and $b$, i.e., say $g_{a_2} = g_{b_2} = g_{ab}$, whereas, the other two are example specific, i.e., $\langle g_{a_1}, g_{b_1} \rangle = 0$. Now, the overall gradient is

$$g = g_a + g_b = g_{a_1} + 2\, g_{ab} + g_{b_1}.$$

Observe that the gradient is stronger in the direction that simultaneously helps both examples and thus the corresponding parameter changes are bigger than those those that only benefit only one example.[1]

It is important to emphasize that the notion of similarity used above (i.e., which examples are considered similar) is not a constant but changes in the course of training as network parameters change. It starts from a mostly task independent notion due to random initialization and is bootstrapped in the course of training to be task dependent. We say "mostly" because even with random initialization, examples that are syntactically close are treated similarly (e.g., two images differing in the intensities of some pixels as opposed to two images where one is a translated version of the other).

The relationship between strong gradients and generalization can also be understood through the lens of algorithmic stability (Bousquet & Elisseeff, 2002): strong gradient directions are more stable since the presence or absence of a single example does not impact them as much, as opposed to weak gradient directions which may altogether disappear if a specific example is missing from the training set. With this observation, we can reason inductively about the stability of GD: since the initial values of the parameters do not depend on the training data, the initial function mapping examples to their gradients is stable. Now, if all parameter updates are due to strong gradient directions, then stability is preserved. However, if some parameter updates are due to weak gradient directions, then stability is diminished. Since stability (suitably formalized) is equivalent to generalization (Shalev-Shwartz et al., 2010), this allows us to see how generalization may degrade as training progresses. Based on this insight, we shall see later how a simple modification to GD to suppress the weak gradient directions can dramatically reduce overfitting.

In addition to providing insight into why GD generalizes in practice, we believe that the Coherent Gradients hypothesis can help explain several other empirical observations about deep learning in the literature:

(a) Learning is slower with random labels than with real labels (Zhang et al., 2017; Arpit et al., 2017)

(b) Robustness to large amounts of label noise (Rolnick et al., 2017)

(c) Early stopping leads to better generalization (Caruana et al., 2000)

(d) Increasing capacity improves generalization (Caruana et al., 2000; Neyshabur et al., 2018)

(e) The existence of adversarial initialization schemes (Liu et al., 2019)

(f) GD detects common patterns even when trained with random labels (Chatterjee & Mishchenko, 2019)

---

[1] While the mechanism is easiest to see with full or large minibatches, we believe it holds even for small minibatches (though there one has to consider the bias in updates over time).

A direct experimental verification of the Coherent Gradients hypothesis is challenging since the notion of similarity between examples depends on the parameters of the network and thus changes during training. Our approach, therefore, is to design intervention experiments where we establish a baseline and compare it against variants designed to test some aspect or prediction of the theory. As part of these experiments, we replicate the observations (a)–(c) in the literature noted above, and analyze the corresponding explanations provided by Coherent Gradients (§2), and outline for future work how (d)–(f) may be accounted for (§5).

In this paper, we limit our study to simple baselines: vanilla Stochastic Gradient Descent (SGD) on MNIST using fully connected networks. We believe that this is a good starting point, since even in this simple setting, with all frills eliminated (e.g., inductive bias from architecture or explicit regularization, or a more sophisticated optimization procedure), we are challenged to find a satisfactory explanation of why SGD generalizes well. Furthermore, our prior is that the difference between weak and strong directions is small at any one step of training, and therefore having a strong learning signal as in the case of MNIST makes a direct analysis of gradients easier. It also has the benefit of having a smaller carbon footprint and being easier to reproduce. Finally, based on preliminary experiments on other architectures and datasets we are optimistic that the insights we get from studying this simple setup apply more broadly.

## 2   EFFECT OF REDUCING SIMILARITY BETWEEN EXAMPLES

Our first test of the Coherent Gradients hypothesis is to see what happens when we reduce similarity between examples. Although, at any point during training, we do not know which examples are similar, and which are different, we can (with high probability) reduce the similarity among training examples simply by injecting label noise. In other words, under any notion of similarity, adding label noise to a dataset that has clean labels is likely to make similar examples less similar. Note that this perturbation does not reduce coherence since gradients still depend on the examples. (To break coherence, we would have to make the gradients independent of the training examples which would requiring perturbing SGD itself and not just the dataset).

### 2.1   SETUP

For our baseline, we use the standard MNIST dataset of 60,000 training examples and 10,000 test examples. Each example is a 28x28 pixel grayscale handwritten digit along with a label ('0'–'9'). We train a fully connected network on this dataset. The network has one hidden layer with 2048 ReLUs and an output layer with a 10-way softmax. We initialize it with Xavier and train using vanilla SGD (i.e., no momentum) using cross entropy loss with a constant learning rate of 0.1 and a minibatch size of 100 for $10^5$ steps (i.e., about 170 epochs). We do not use any explicit regularizers.

We perturb the baseline by modifying *only* the dataset and keeping all other aspects of the architecture and learning algorithm fixed. The dataset is modified by adding various amounts of noise (25%, 50%, 75%, and 100%) to the labels of the training set (but not the test set). This noise is added by taking, say in the case of 25% label noise, 25% of the examples at random and randomly permuting their labels. Thus, when we add 25% label noise, we still expect about 75% + 0.1 * 25%, i.e., 77.5% of the examples to have unchanged (i.e. "correct") labels which we call the *proper accuracy* of the modified dataset. In what follows, we call examples with unchanged labels, *pristine*, and the remaining, *corrupt*. Also, from this perspective, it is convenient to refer to the original MNIST dataset as having 0% label noise.

We use a fully connected architecture instead of a convolutional one to mitigate concerns that some of the difference in generalization between the original MNIST and the noisy variants could stem from architectural inductive bias. We restrict ourselves to only 1 hidden layer to have the gradients be as well-behaved as possible. Finally, the network width, learning rate, and the number of training steps are chosen to ensure that exactly the same procedure is usually able to fit all 5 variants to 100% training accuracy.

## 2.2 QUALITATIVE PREDICTIONS

Before looking at the experimental results, it is useful to consider what Coherent Gradients can qualitatively say about this setup. In going from 0% label noise to 100% label noise, as per experiment design, we expect examples in the training set to become more dissimilar (no matter what the current notion of similarity is). Therefore, we expect the per-example gradients to be less aligned with each other. This in turn causes the overall gradient to become more diffuse, i.e., stronger directions become relatively weaker, and consequently, we expect it to take longer to reach a given level of accuracy as label noise increases, i.e., to have a lower *realized learning rate*.

This can be made more precise by considering the following heuristic argument. Let $\theta_t$ be the vector of trainable parameters of the network at training step $t$. Let $\mathcal{L}$ denote the loss function of the network (over all training examples). Let $g_t$ be the gradient of $\mathcal{L}$ at $\theta_t$ and let $\alpha$ denote the learning rate. By Taylor expansion, to first order, the change $\Delta\mathcal{L}_t$ in the loss function due to a small gradient descent step $h_t = -\alpha \cdot g_t$ is given by

$$\Delta\mathcal{L}_t := \mathcal{L}(\theta_t + h_t) - \mathcal{L}(\theta_t) \approx \langle g_t, h_t \rangle = -\alpha \cdot \langle g_t, g_t \rangle = -\alpha \cdot \|g_t\|^2 \tag{1}$$

where $\|\cdot\|$ denotes the $l_2$-norm. Now, let $g_{te}$ denote the gradient of training example $e$ at step $t$. Since the overall gradient is the sum of the per-example gradients, we have,

$$\|g_t\|^2 = \langle g_t, g_t \rangle = \langle \sum_e g_{te}, \sum_e g_{te} \rangle = \sum_{e,e'} \langle g_{te}, g_{te'} \rangle = \sum_e \|g_{te}\|^2 + \sum_{\substack{e,e' \\ e \neq e'}} \langle g_{te}, g_{te'} \rangle \tag{2}$$

Now, heuristically, let us assume that all the $\|g_{te}\|$ are roughly the same and equal to $\|g_t^\circ\|$ which is not entirely unreasonable (at least at the start of training, if the network has no *a priori* reason to treat different examples very differently). If all the per-example gradients are approximately orthogonal (i.e., $\langle g_{te}, g_{te'} \rangle \approx 0$ for $e \neq e'$), then $\|g_t\|^2 \approx m \cdot \|g_t^\circ\|^2$ where $m$ is the number of examples. On the other hand, if they are approximately the same (i.e., $\langle g_{te}, g_{te'} \rangle \approx \|g_t^\circ\|^2$), then $\|g_t\|^2 \approx m^2 \cdot \|g_t^\circ\|^2$. Thus, we expect that greater the agreement in per-example gradients, the faster loss should decrease.

Finally, for datasets that have a significant fractions of pristine and corrupt examples (i.e., the 25%, 50%, and 75% noise) we can make a more nuanced prediction. Since, in those datasets, the pristine examples as a group are still more similar than the corrupt ones, we expect the pristine gradients to continue to align well and sum up to a strong gradient. Therefore, we expect them to be learned faster than the corrupt examples, and at a rate closer to the realized learning rate in the 0% label noise case. Likewise, we expect the realized learning rate on the corrupt examples to be closer to the 100% label noise case. Finally, as the proportion of pristine examples falls with increasing noise, we expect the realized learning rate for pristine examples to degrade.

Note that this provides an explanation for the observation in the literature that that networks can learn even when the number of examples with noisy labels greatly outnumber the clean examples *as long as the number of clean examples is sufficiently large* (Rolnick et al., 2017) since with too few clean examples the pristine gradients are not strong enough to dominate.

## 2.3 AGREEMENT WITH EXPERIMENT

Figure 1(a) and (b) show the training and test curves for the baseline and the 4 variants. We note that for all 5 variants, at the end of training, we achieve 100% training accuracy but different amounts of generalization. As expected, SGD is able to fit random labels, yet when trained on real data, generalizes well. Figure 1(c) shows the reduction in training loss over the course of training, and Figure 1(d) shows the fraction of pristine and corrupt labels learned as training processes.

The results are in agreement with the qualitative predictions made above:

1. In general, as noise increases, the time taken to reach a given level of accuracy (i.e., realized learning rate) increases.

2. Pristine examples are learned faster than corrupt examples. They are learned at a rate closer to the 0% label noise rate whereas the corrupt examples are learned at a rate closer to the 100% label noise rate.

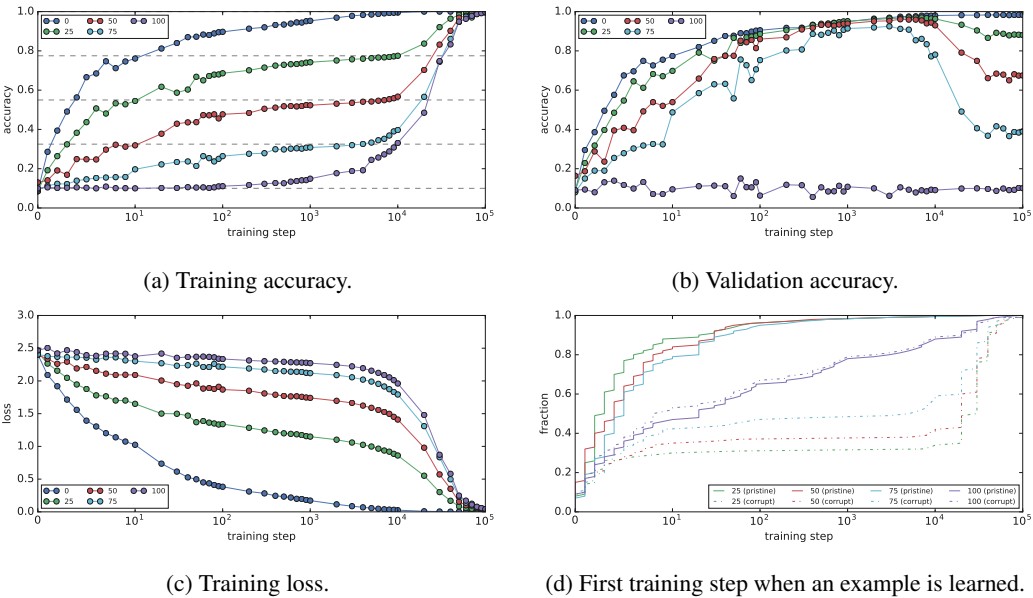

(a) Training accuracy.

(b) Validation accuracy.

(c) Training loss.

(d) First training step when an example is learned.

Figure 1: Results of the experiment to reduce similarity by adding label noise (§2).

3. With fewer pristine examples, their learning rate reduces. This is most clearly seen in the first few steps of training by comparing say 0% noise with 25% noise.

Using Equation 1, note that the magnitude of the slope of the training loss curve is a good measure of the square of the $l_2$-norm of the overall gradient. Therefore, from the loss curves of Figure 1(c), it is clear that in early training, the more the noise, the weaker the $l_2$-norm of the gradient. If we assume that the per-example $l_2$-norm is the same in all variants at start of training, then from Equation 2, it is clear that with greater noise, the gradients are more dissimilar.

Finally, we note that this experiment is an instance where early stopping (e.g., Caruana et al. (2000)) is effective. Coherent gradients and the discussion in §2.2 provide some insight into this: Strong gradients both generalize well (they are stable since they are supported by many examples) and they bring the training loss down quickly for those examples. Thus early stopping maximizes the use of strong gradients and limits the impact of weak gradients. (The experiment in the §3 discusses a different way to limit the impact of weak gradients and is an interesting point of comparison with early stopping.)

## 2.4 Analyzing Strong and Weak Gradients

Within each noisy dataset, we expect the pristine examples to be more similar to each other and the corrupt ones to be less similar. In turn, based on the training curves (particularly, Figure 1 (d)), during the initial part of training, this should mean that the gradients from the pristine examples should be stronger than the gradients from the corrupt examples. We can study this effect via a different decomposition of square of the $l_2$-norm of the gradient (of equivalently upto a constant, the change in the loss function):

$$\langle g_t, g_t \rangle = \langle g_t, g_t^p + g_t^c \rangle = \langle g_t, g_t^p \rangle + \langle g_t, g_t^c \rangle$$

where $g_t^p$ and $g_t^c$ are the sum of the gradients of the pristine examples and corrupt examples respectively. (We cannot decompose the overall norm into a sum of norms of pristine and corrupt due to the cross terms $\langle g_t^p, g_t^c \rangle$. With this decomposition, we attribute the cross terms equally to both.) Now, set $f_t^p = \frac{\langle g_t, g_t^p \rangle}{<g_t, g_t>}$ and $f_t^c = \frac{\langle g_t, g_t^c \rangle}{\langle g_t, g_t \rangle}$. Thus, $f_t^p$ and $f_t^c$ represent the fraction of the loss reduction due to pristine and corrupt at each time step respectively (and we have $f_t^p + f_t^c = 1$), and based on the foregoing, we expect the pristine fraction to be a larger fraction of the total when training starts and to diminish as training progresses and the pristine examples are fitted.

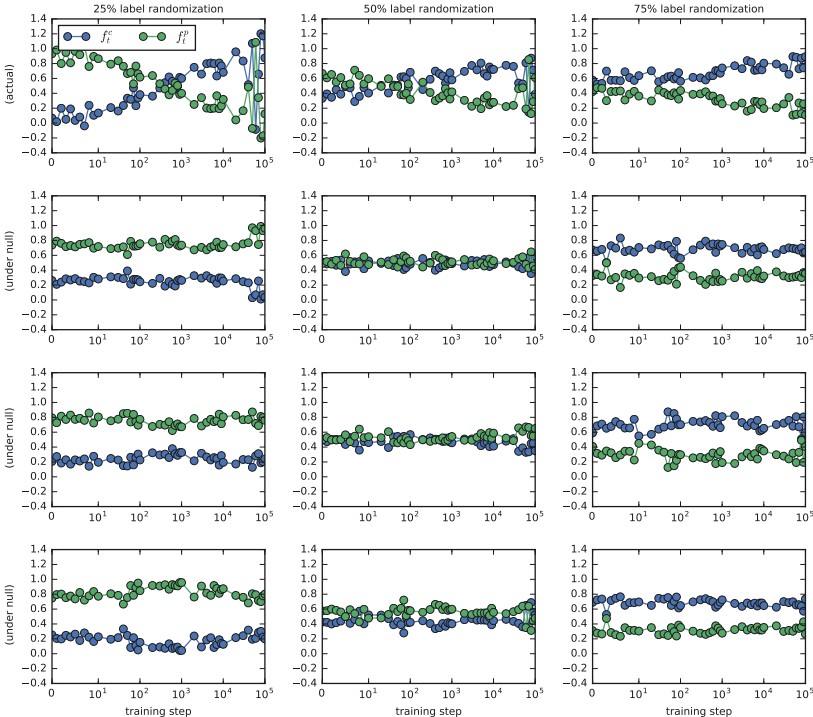

Figure 2: Relative contributions of pristine (similar) and corrupt (dissimilar) examples to point-in-time loss reduction. To get a sense of statistical significance, we show the actual statistic as well as 3 simulations under the null assuming there is no difference. See §2.4.

The first row of Figure 2 shows a plot of estimates of $f_t^p$ and $f_t^c$ for 25%, 50% and 75% noise. These quantities were estimated by recording a sample of 400 per-example gradients for 600 weights (300 from each layer) in the network. We see that for 25% and 50% label noise, $f_t^p$ initially starts off higher than $f_t^c$ and after a few steps they cross over. This happens because at that point all the pristine examples have been fitted and for most of the rest of training the corrupt examples need to be fitted and so they largely contribute to the $l_2$-norm of the gradient (or equivalently by Equation 1 to loss reduction). Only at the end when the corrupt examples have also been fit, the two curves reach parity. In the case of 75% noise, we see that the cross over doesn't happen, but there is a slight slope downwards for the contribution from pristine examples. We believe this is because of the sheer number of corrupt examples, and so even though the individual corrupt example gradients are weak, their sum dominates.

To get a sense of statistical significance in our hypothesis that there is a difference between the pristine and corrupt examples as a group, in the remaining rows of Figure 2, we construct a null world where there is no difference between pristine and corrupt. We do that by randomly permuting the "corrupt" and "pristine" designations among the examples (instead of using the actual designations) and replotting. Although the null pristine and corrupt curves are mirror images (as they must be even in the null world since each example is given one of the two designations), we note that for 25% and 50% they do not cross over as they do with the real data. This increases our confidence that the null may be rejected. The 75% case is weaker but only the real data shows the slight downward slope in pristine which none of the nulls typically show. However, all the nulls do show that corrupt is more than pristine which increases our confidence that this is due to the significantly differing sizes of the two sets. (Note that this happens in reverse in the 25% case: pristine is always above corrupt, but they never cross over in the null worlds.)

To get a stronger signal for the difference between pristine and corrupt in the 75% case, we can look at a different statistic that adjusts for the different sizes of the pristine and corrupt sets. Let $|p|$ and $|c|$

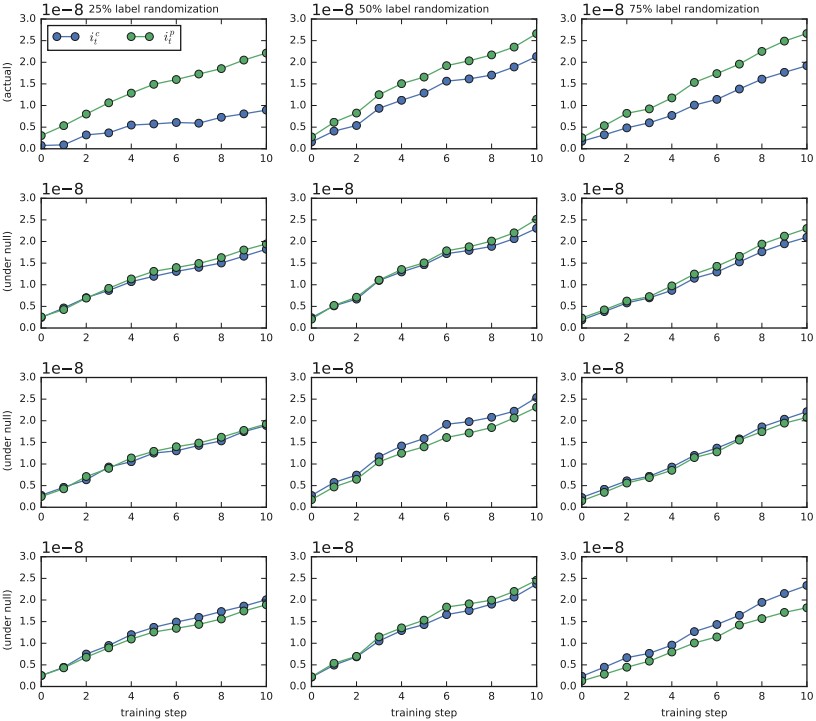

Figure 3: Contributions of mean pristine (similar) and corrupt (dissimilar) examples to loss reduction accumulated over first few steps of training. To get a sense of statistical significance, we show the actual statistic as well as 3 simulations under the null assuming there is no difference. See §2.4.

be the number of pristine and corrupt examples respectively. Define

$$i_t^p := \frac{1}{|p|} \sum_{t'=0}^{t} \langle g_{t'}, g_{t'}^p \rangle \quad \text{and} \quad i_t^c := \frac{1}{|c|} \sum_{t'=0}^{t} \langle g_{t'}, g_{t'}^c \rangle$$

which represents to a first order and upto a scale factor ($\alpha$) the mean cumulative contribution of a pristine or corrupt example up until that point in training (since the total change in loss from the start of training to time $t$ is approximately the sum of first order changes in the loss at each time step).

The first row of Figure 3 shows $i_t^p$ and $i_t^c$ for the first 10 steps of training where the difference between pristine and corrupt is the most pronounced. As before, to give a sense of statistical significance, the remaining rows show the same plots in null worlds where we randomly permute the pristine or corrupt designations of the examples. The results appear somewhat significant but not overwhelmingly so. It would be interesting to redo this on the entire population of examples and trainable parameters instead of a small sample.

## 3 EFFECT OF SUPPRESSING WEAK GRADIENT DIRECTIONS

In the second test of the Coherent Gradients hypothesis, we change GD itself in a very specific (and to our knowledge, novel) manner suggested by the theory. Our inspiration comes from random forests. As noted in the introduction, by building sufficiently deep trees a random forest algorithm can get perfect training accuracy with random labels, yet generalize well when trained on real data. However, if we limit the tree construction algorithm to have a certain minimum number of examples in each leaf, then it no longer overfits. In the case of GD, we can do something similar by suppressing the weak gradient directions.

## 3.1 SETUP

Our baseline setup is the same as before (§2.1) but we add a new dimension by modifying SGD to update each parameter with a "winsorized" gradient where we clip the most extreme values (outliers) among all the per-example gradients. Formally, let $g_{we}$ be the gradient for the trainable parameter $w$ for example $e$. The usual gradient computation for $w$ is

$$g_w = \sum_e g_{we}$$

Now let $c \in [0, 50]$ be a hyperparameter that controls the level of winsorization. Define $l_w$ to be the $c$-th percentile of $g_{we}$ taken over the examples. Similarly, let $u_w$ be the $(100 - c)$-th percentile. Now, compute the $c$-winsorized gradient for $w$ (denoted by $g_w^c$) as follows:

$$g_w^c := \sum_e \text{clip}(g_{we}, l_w, u_w)$$

The change to gradient descent is to simply use $g_w^c$ instead of $g_w$ when updating $w$ at each step.

Note that although this is conceptually a simple change, it is computationally very expensive due to the need for per-example gradients. To reduce the computational cost we only use the examples in the minibatch to compute $l_w$ and $u_w$. Furthermore, instead of using 1 hidden layer of 2048 ReLUs, we use a smaller network with 3 hidden layers of 256 ReLUs each, and train for 60,000 steps (i.e., 100 epochs) with a fixed learning rate of 0.1. We train on the baseline dataset and the 4 noisy variants with $c \in \{0, 1, 2, 4, 8\}$. Since we have 100 examples in each minibatch, the value of $c$ immediately tells us how many outliers are clipped in each minibatch. For example, $c = 2$ means the 2 largest and 2 lowest values of the per-example gradient are clipped (independently for each trainable parameter in the network), and $c = 0$ corresponds to unmodified SGD.

## 3.2 QUALITATIVE PREDICTIONS

If the Coherent Gradient hypothesis is right, then the strong gradients are responsible for making changes to the network that generalize well since they improve many examples simultaneously. On the other hand, the weak gradients lead to overfitting since they only improve a few examples. By winsorizing each coordinate, we suppress the most extreme values and thus ensure that a parameter is only updated in a manner that benefits multiple examples. Therefore:

- Since $c$ controls which examples are considered extreme, the larger $c$ is, the less we expect the network to overfit.
- But this also makes it harder for the network to fit the training data, and so we expect the training accuracy to fall as well.
- Winsorization will not completely eliminate the weak directions. For example, for small values of $c$ we should still expect overfitting to happen over time though at a reduced rate since only the most egregious outliers are suppressed.

## 3.3 AGREEMENT WITH EXPERIMENT

The resulting training and test curves shown in Figure 4. The columns correspond to different amounts of label noise and the rows to different amounts of winsorization. In addition to the training and test accuracies (ta and va, respectively), we show the level of overfit which is defined as $\text{ta} - [\epsilon \cdot \frac{1}{10} + (1 - \epsilon) \cdot \text{va}]$ to account for the fact that the test labels are not randomized.

We see that the experimental results are in agreement with the predictions above. In particular,

- For $c > 1$, training accuracies do not exceed the proper accuracy of the dataset, though they may fall short, specially for large values of $c$.
- The rate at which the overfit curve grows goes down with increasing $c$.

Additionally, we notice that with a large amount of winsorization, the training and test accuracies reach a maximum and then go down. Part of the reason is that as a result of winsorization, each step is no longer in a descent direction, i.e., this is no longer gradient descent.

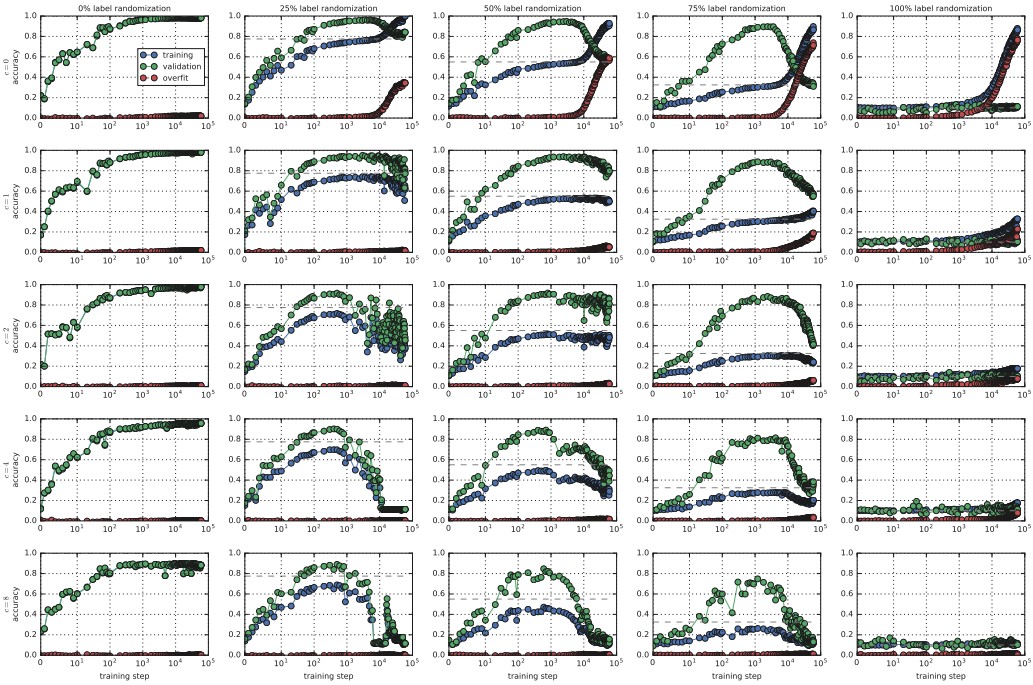

Figure 4: Effect of suppressing weak gradient directions by eliminating outlier per-example gradients. This is done independently for each trainable parameter. Overfit is measured after accounting for the fact that test labels are not randomized (§3.3).

## 4 DISCUSSION AND RELATED WORK

Although there has been a lot of work in recent years in trying to understand generalization in Deep Learning, no entirely satisfactory explanation has emerged so far.

There is a rich literature on aspects of the stochastic optimization problem such as the loss landscape and minima (e.g., Choromanska et al. (2015); Zhu et al. (2018)), the curvature around stationary points (e.g., Hochreiter & Schmidhuber (1997); Keskar et al. (2016); Dinh et al. (2017); Wu et al. (2018)), and the implications of stochasticity due to sampling in SGD (e.g., Simsekli et al. (2019)). However, we believe it should be possible to understand generalization without a detailed understanding of the optimization landscape. For example, since stopping early typically leads to small generalization gap, the nature of the solutions of GD (e.g., stationary points, the limit cycles of SGD at equilibrium) cannot be solely responsible for generalization. In fact, from this observation, it would appear that an inductive argument for generalization would be more natural. Likewise, there is reason to believe that stochasticity is not fundamental to generalization (though it may help). For example, modifying the experiment in §2.1 to use full batch leads to similar qualitative generalization results. This is consistent with other small scale studies (e.g., Figure 1 of Wu et al. (2018)) though we are not aware of any large scale studies on full batch.

Our view of optimization is a simple, almost combinatorial, one: gradient descent is a greedy search with some hill-climbing thrown in (due to sampling in SGD and finite step size). Therefore, we worry less about the quality of solutions reached, but more about staying "feasible" at all times during the search. In our context, feasibility means being able to generalize; and this naturally leads us to look at the transition dynamics to see if that preserves generalizability.

Another approach to understanding generalization, is to argue that gradient-based optimization induces a form of implicit regularization leading to a bias towards models of low complexity. This is an extension of the classical approach where bounding a complexity measure leads to bounds on the generalization gap. As is well known, classical measures of complexity (also called capacity) do not work well. For example, sometimes adding more parameters to a net can help generalization (see for e.g. Lawrence et al. (1996); Neyshabur et al. (2018)) and, as we have seen, VC-Dimension

and Rademacher Complexity-based bounds must be vacuous since networks can memorize random labels and yet generalize on real data. This has led to a lot of recent work in identifying better measures of complexity such as spectrally-normalized margin (Bartlett et al., 2017), path-based group norm (Neyshabur et al., 2018), a compression-based approach (Arora et al., 2018), etc. However, to our knowledge, none of these measures is entirely satisfactory for accounting for generalization in practice. Please see Nagarajan & Kolter (2019) for an excellent discussion of the challenges.

We rely on a different classical notion to argue generalization: algorithmic stability (see Bousquet & Elisseeff (2002) for a historical overview). We have provided only an informal argument in Section 1, but there has been prior work by Hardt et al. (2016) in looking at GD and SGD through the lens of stability, but their formal results do not explain generalization in practical settings (e.g., multiple epochs of training and non-convex objectives). In fact, such an attempt appears unlikely to work since our experimental results imply that any stability bounds for SGD that do not account for the actual training data must be vacuous! (This was also noted by Zhang et al. (2017).) That said, we believe stability is the right way to think about generalization in GD for a few reasons. First, since by Shalev-Shwartz et al. (2010) stability, suitably formalized, is equivalent to generalization. Therefore, in principle, any explanation of generalizability for a learning problem must—to borrow a term from category theory—factor through stability. Second, a stability based analysis may be more amenable to taking the actual training data into account (perhaps by using a "stability accountant" similar to a privacy accountant) which appears necessary to get non-vacuous bounds for practical networks and datasets. Finally, as we have seen with the modification in §3, a stability based approach is not just descriptive but prescriptive[2] and can point the way to better learning algorithms.

Finally, we look at two relevant lines of work pointed out by a reviewer. First, Rahaman et al. (2019) compute the Fourier spectrum of ReLU networks and argue based on heuristics and experiments that these networks learn low frequency functions first. In contrast, we focus not on the function learnt, but on the mechanism in GD to detect commonality. This leads to a perspective that is at once simpler and more general (for e.g., it applies equally to networks with other activation functions, with attention, LSTMs, and discrete (combinatorial) inputs). Furthermore, it opens up a path to analyzing generalization via stability. It is is not clear if Rahaman et al. (2019) claim a causal mechanism, but their analysis does not suggest an obvious intervention experiment such as ours of §3 to test causality. There are other experimental results that show biases towards linear functions (Nakkiran et al., 2019) and functions with low descriptive complexity (Valle-Perez et al., 2019) but these papers do not posit a causal mechanism. It is interesting to consider if Coherent Gradients can provide a unified explanation for these observed biases.

Second, Fort et al. (2019) propose a descriptive statistic *stiffness* based on pairwise per-example gradients and show experimentally that it can be used to characterize generalization. Sankararaman et al. (2019) propose a very similar statistic called *gradient confusion* but use it to study the speed of training. Unlike our work, these do not propose causal mechanisms for generalization, but these statistics (which are different from those in §2.4) could be useful for the further study of Coherent Gradients.

## 5 DIRECTIONS FOR FUTURE WORK

Does the Coherent Gradients hypothesis hold in other settings such as BERT, ResNet, etc.? For that we would need to develop more computationally efficient tests. Can we use the state of the network to explicitly characterize which examples are considered similar and study this evolution in the course of training? We expect non-parametric methods for similarity such as those developed in Chatterjee & Mishchenko (2019) and their characterization of "easy" examples (i.e., examples learnt early as per Arpit et al. (2017)) as those with many others like them, to be useful in this context.

Can Coherent Gradients explain adversarial initializations (Liu et al., 2019)? The adversarial initial state makes semantically similar examples purposefully look different. Therefore, during training, they continue to be treated differently (i.e., their gradients share less in common than they would if starting from a random initialization). Thus, fitting is more case-by-case and while it achieves good final training accuracy, it does not generalize.

---

[2]See https://www.offconvex.org/2017/12/08/generalization1/ for a nice discussion of the difference.

Can Coherent Gradients along with the Lottery Ticket Hypothesis (Frankle & Carbin, 2018) explain the observation in Neyshabur et al. (2018) that wider networks generalize better? By Lottery Ticket, wider networks provide more chances to find initial gradient directions that improve many examples, and by Coherent Gradients, these popular hypotheses are learned preferentially (faster).

Can we use the ideas behind Winsorized SGD from §3 to develop a computationally efficient learning algorithm with generalization (and even privacy) guarantees? How does winsorized gradients compare in practice to the algorithm proposed in Abadi et al. (2016) for privacy? Last, but not least, can we use the insights from this work to design learning algorithms that operate natively on discrete networks?

## ACKNOWLEDGMENTS

I thank Alan Mishchenko, Shankar Krishnan, Piotr Zielinski, Chandramouli Kashyap, Sergey Ioffe, Michele Covell, and Jay Yagnik for helpful discussions.

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
