# OpenReview forum: "Coherent Gradients: An Approach to Understanding Generalization in Gradient Descent-based Optimization"
_ICLR.cc/2020/Conference — Accept (Poster)_

### Official Review · AnonReviewer3 · 2019-10-20
**Official Blind Review #3**

**Rating:** 3

**Review:**


The paper studies the link between alignment of the gradients computed on different examples, and generalization of deep neural networks. The paper tackles an important research question, is very clearly written, and proposes an insightful metric. In particular, through the lenses of the metric it is possible to understand better the learning dynamics on random labels. However, the submission seems to have limited novelty, based on which I am leaning towards rejecting the paper.

Detailed comments

1. The prior and concurrent work is not discussed sufficiently:

a) The novelty of the "Coherent Gradients hypothesis" is not clear to me. First, the empirical fact that some examples are easier to learn than others in training of deep networks was the key focus of [5].

Hence, "Coherent Graident Hypothesis" should be mostly considered an explanation for why simple examples are/simple function are learned first. "Coherent Gradient Hypothesis" proposes that the key mechanism behind this phenomena is that simple examples/functions have co-aligned gradients and hence a larger "effective" learning rate. However, there are already quite convincing and closely related hypotheses. For example, the spectral bias interpretation of deep networks [2] and (2) suggests the same view actually. Just expressed in a different formalism, but can be also casted as having a higher effective learning rate for the strongest modes. Similarly, [3] proposes that SGD learns functions of increased complexity. A detailed comparison between these hypotheses is needed.

b) "Gradient coherence" metric is very closely related to Stiffness studied in [1] (01.2019 on arXiv). [1] studies the cosine (or sign) between gradients coming from different examples, and reach quite similar conclusions. It is also worth noting that [6, 7] propose and study a very similar metric as well. While arXiv submissions is not consider prior work, these three preprints should be discussed in detail in the submission.

c) It should be also remarked that "Coherent Gradient hypothesis" is to some extend folk knowledge. In particular, it is quite well known and also brought to the attention of the deep learning community that in linear regression strongest modes of the datasets as learned first when training using GD (see for instance [4]), which causally speaking stems directly from gradient coherence; these modes correspond to the largest eigenvalues of the (constant) Hessian. To make it more precise: consider that GD solving linear regression can be seen as having higher "effective" learning rates along the strongest modes in the dataset.

2. Experiments on random labels and restricting gradient norms are interesting. However, [5] should be cited. They experimented with regularization impact on memorization, which due to the addition of noise, probably also supresses weak gradients.

3. Experiments on MNIST do not feel adequate. While I do not doubt the validity of the experimental results, the paper should include results on another dataset; ideally from other domain than vision.

4. Plots in Figure 4 are too small to read. I would recommend moving half of them to the Supplement?

5. "Understanding why solutions of the optimization problem on the training sample carry over to the population at large" - Not sure what do you mean here. Could you please clarify?

6. "Furthermore, while SGD is critical for computational speed, from our experiments and others (Keskar et al., 2016; Wu et al., 2017; Zhang et al., 2017) it appears not to be necessary.". Please note there is very little work on training with GD large models. Also, citing in this context Keskar is misleading. Wasn't the whole point of Keskar to show why large batch size training overfits? Finally, there are many papers on studying the role of learning rate and batch size in generalization (not computational speed). I think this sentence should be rewritten to clarify what is the experimental data that GD is "sufficient", and SGD is just needed for "computational speed".

References

[1] Stanislav Fort et al, Stiffness: A New Perspective on Generalization in Neural Networks, https://arxiv.org/abs/1901.09491
[2] Rahaman et al, On the Spectral Bias of Neural Networks, https://arxiv.org/abs/1806.08734
[3] Nakkiran et al, SGD on Neural Networks Learns Functions of Increasing Complexity, https://arxiv.org/abs/1905.11604
[4] Goh, Why Momentum Really Works, https://distill.pub/2017/momentum/
[5] Arpit et al, A Closer Look at Memorization in Deep Networks, https://arxiv.org/abs/1706.05394
[6] He and Su, The Local Elasticity of Neural Networks, https://arxiv.org/abs/1910.06943
[7] Sankararaman, The Impact of Neural Network Overparameterization on Gradient Confusion and Stochastic Gradient Descent, https://arxiv.org/abs/1904.06963

**Experience Assessment:**

I have published one or two papers in this area.

**Review Assessment: Checking Correctness Of Derivations And Theory:**

I carefully checked the derivations and theory.

**Review Assessment: Checking Correctness Of Experiments:**

I carefully checked the experiments.

**Review Assessment: Thoroughness In Paper Reading:**

I read the paper thoroughly.

---

> ### Author Response · Authors · 2019-11-10
> **Response to Detailed Comments**
>
> Please see our overall response first. (The comment numbering below corresponds to the numbering in the original review.)
>
> 1 (a) (b). We believe we have addressed this in our comments above but please let us know if not.
>
> 1 (c). We are not sure what you mean by “mode” in the context of [4]. The word does not appear in [4] which is a tutorial on why momentum works. (To clarify for other readers, we do not look at momentum or other optimizers in our paper focussing only on vanilla GD.)
>
> It is not always easy to explain generalization in linear models either. For instance see discussion in Section 5 in https://arxiv.org/abs/1611.03530 on precisely this topic.
>
> We would love to include a discussion of the folklore, but we and our immediate network of colleagues are not familiar with this. Also we would expect [2, 3] to also reference such folk knowledge, so maybe there’s something we can point to there?
>
> 2. We cite [5] (see Section 5, first para). Also see discussion above on simple examples above.
>
> 3. This is fair. Our thinking is that even in the MNIST case we do not understand why generalization happens, so why not explain that first. Furthermore, even on MNIST experiments can computationally expensive (e.g. our expt in Section 3 as well as full-batch gradient). We hope this work inspires others to study this on other architectures, optimizers, datasets.
>
> 4. We’ll consider this. We thought having the 5x5 grid makes it easier to get the qualitative big picture. On the screen, the PDF should be zoomable to see full detail but do let us know if that is not your experience.
>
> 5. We are referring to generalization proper. We’ll rephrase as “Understanding generalization proper, i.e., ..”.
>
> 6. Yes, we should drop the Keskar reference since they do not train with full batch. However, do note that even their large batch models generalize. The experiment in Wu et al. 2017 (e.g. Figure 1) shows that (full batch) gradient descent generalizes well (albeit not as well as stochastic gradient descent). We find the same thing in our experiments with full batch (one benefit of restricting our focus to MNIST) -- see our response to Reviewer #2 for the data. Thus we know that at least in these cases, stochasticity is not fundamentally responsible for generalization. Also note in this context that the recent large scale study (https://arxiv.org/abs/1811.03600) found no evidence that larger batch sizes degrade out-of-sample performance.
>
> We will rephrase to say that based on our experiments we believe that stochasticity is not fundamental to generalization (though may help with it) and that his has also been found in other experiments in the literature such as [Wu et al. 2017 (Figure 1)]. Furthermore, this is consistent with what is known about large batches [https://arxiv.org/abs/1811.03600].
>
> Please let us know if this does not sound fair.

---

> > ### Comment · AnonReviewer3 · 2019-11-11
> > **Batch-size is just one of a few somewhat equivalent knobs that influence generalization.**
> >
> > Ad. 1-5. Thanks for the answers.
> >
> > Ad 6. The papers you mention establish equivalence between the regularization effect of small batch-size and other regularizers. It is not that they turn on a large batch size, and it just works. For additional experimental data on this equivalence see papers on learning rate, see for learning rate https://arxiv.org/abs/1711.00489 and https://arxiv.org/abs/1711.04623, and weight decay https://arxiv.org/abs/1810.12281. I think implicit and explicit regularizations due to learning rate, batch size, weight decay, and others are quite important. I actually do not think there is any study on training a realistic model with just GD (and without adapting any of the above).
> >
> > In short, I am OK with saying stochasticity is not needed, as long as you add that if stochasticity is removed, other implicit or explicit regularizers are needed to achieve good generalization.

---

> > > ### Author Response · Authors · 2019-11-11
> > > **Yes, but is stochasticity fundamentally required?**
> > >
> > > Thank you. We are aware of the equivalence and the work in this area but we are after something a little different.
> > >
> > > One of the benefits of focussing on MNIST (a simple case where we still don't have a fully satisfactory explanation of generalization), we can actually do a full batch v/s stochastic comparison keeping all else same. We (like Wu et al.) find no evidence that full batch doesn't generalize. So at least in the concrete case we are studying, stochasticity does not appear to be necessary for generalization.
> > >
> > > But a question for you: Note that we are not particular interested here in a practical result. So with full GD we can train for same number of training steps as SGD (so the total compute would be far more than with SGD). This is quite different from the focus of the papers you in refer to.
> > >
> > > Now, in this setting, do you know of any study where just gradient descent (instead of SGD) completely fails to generalize on a real dataset like ImageNet? In other words, an experiment that shows stochasticity is necessary for generalization? Just as in https://arxiv.org/abs/1611.03530, where they found explicit regularizers help (i.e. improve) but are not fundamental to generalization, we believe the same about stochasticity and would love to know of a counter example.
> > >
> > > If not (and perhaps that's what you are saying), perhaps that is a very nice experiment.
> > >
> > > Please let us know. At any rate, we will put in the more nuanced discussion in the revised paper.

---

> > > > ### Comment · AnonReviewer3 · 2019-11-11
> > > > **Feels a bit like a strawman argument**
> > > >
> > > > First of all, thanks for agreeing to add the nuanced discussion.
> > > >
> > > > Second, I have an issue with saying that the fact deep networks do not fully memorize data is surprising. To rephrase what you are saying: the interesting thing is that when running GD on a deep network it doesn't memorize the data because there exists a hypothesis in the hypothesis class that would memorize, but GD doesn't pick it.
> > > >
> > > > However, we already know (analytically speaking) why in the case of linear regression GD doesn't pick the memorizing hypothesis (even if it is in the hypothesis class, e.g. by massively overparametrizing the model).
> > > >
> > > > Hence, this feels a bit like a strawman argument to say it is surprising GD doesn't fully memorize data when used to train a deep network.
> > > >
> > > > See also Thomas Dietterich comment to the paper you mentioned https://openreview.net/forum?id=Sy8gdB9xx.

---

> > > > > ### Author Response · Authors · 2019-11-12
> > > > > **Are you saying that generalization in deep nets is well understood?**
> > > > >
> > > > > Thanks for your response but we are confused.
> > > > >
> > > > > When you say it is not surprising that GD doesn't memorize data, are you saying that generalization in deep nets is well understood? But we don't think that is the case. (All the references in our introduction and all the active work in the community seems to indicate otherwise.)
> > > > >
> > > > > [Will abstain from commenting on the review discussion in the other paper since clearly opinions were very divided in that discussion though in the end the committee accepted the paper.]

---

> > > > > > ### Comment · AnonReviewer3 · 2019-11-12
> > > > > > **Deep networks generalize = ?**
> > > > > >
> > > > > > Sorry, if that was confusing. I meant precisely (as I wrote above):
> > > > > >
> > > > > > " To rephrase what you are saying: the interesting thing is that when running GD on a deep network it doesn't memorize the data because there exists a hypothesis in the hypothesis class that would memorize, but GD doesn't pick it."
> > > > > >
> > > > > > In other words, I think it does not seem surprising to me that when training on ImageNet a model, it does not *fully* memorize data, i.e. doesn't get 1/1000 accuracy on the test set.
> > > > > >
> > > > > > If we define "deep networks generalize", as that deep networks achieve surprisingly good generalization compared to other models, then I would fully agree.

---

> > > > > > > ### Author Response · Authors · 2019-11-12
> > > > > > > **Thanks for the clarification.**
> > > > > > >
> > > > > > > .

---

> ### Author Response · Authors · 2019-11-10
> **Response to Overall Comments**
>
> Thank you for your encouragement, for your detailed comments and the additional references. We shall update the paper with a discussion of those references. (We use the reference numbers from your comment in our discussion below.)
>
> In short, the main difference with all the previous works you cite is that our hypothesis is causal as opposed to descriptive. Instead of characterizing the behavior of SGD (in the form of a metric or complexity of functions or examples learnt), we aim at understanding what is the mechanism responsible for generalization.
>
> Our main observation is that the summation of per-example gradients to get the overall gradient amplifies directions (or local moves) that help many examples at the same time. Although simple in retrospect, to our knowledge this perspective has not been studied in the literature and that leads to significant differences with existing work:
>
> (1) Simple Functions. Unlike [2, 3] we do not focus on or claim that simple functions are being learnt first, but instead we focus on commonality being detected across examples (which is a different thing). However, we believe it would be interesting future work to reconcile these two viewpoints. [Also see (4) below.]
>
> (2) Simple Examples. As long as some examples are learnt before others, it is tautological to say there are simple examples and the simple examples have a larger effective learning rate. However, Coherent Gradients sheds light on why (from a SGD perspective) some examples are simple and others aren’t. An example is “simple” if there are many like it (a conjecture made in Section 4 of CM19: https://arxiv.org/abs/1907.01991). See first para of Section 5 where we discuss [5 and CM19]).
>
> (3) A new algorithm, not just metrics. Unlike [1, 6, 7] whose focus is on descriptive metrics to characterize generalization and optimization ease respectively, our causal explanation (which doesn’t appear in any of those papers) enables us to propose and study a natural and fundamental modification to GD (Section 3) to suppress overfitting that does not appear elsewhere in the literature to our knowledge. Finally, our methods and even our descriptive metrics are quite different in the details. Please also see the discussion with authors of [1] above where they seem to have accepted our explanation of differences.
>
> (4) Perspective from Stability. Our perspective (Sections 1 and 4) on why generalization is achieved (via stability of strong gradient directions, and an almost combinatorial view of GD) is at once simpler and more general (e.g. not restricted to classification problems; or just ReLU networks like [2], or to learning problems with continuous x’s (as opposed to discrete x’s, say, as in a language model (since what would Fourier transform on the input domain mean there?))), and as such allows us to see neural nets in the same light as other over-parameterized learning systems such as random forests and nearest neighbors.
>
> In the revised version of the paper, we will discuss the references suggested by you. Our other comments above are amplifications of the points already made in the current writeup, albeit briefly for reasons of page length. We hope that these amplifications help assess novelty of our work and if you still feel we lack novelty we would love to iteratively deepen the discussion with more details from you w.r.t. the related work. We are looking forward to hearing from you on this, before surgically improving the main text to make these points even clearer while remaining within page limits.
>
> (We will respond in a separate comment to the detailed comments.)

---

> > ### Comment · AnonReviewer3 · 2019-11-11
> > **On the novelty**
> >
> > Thank you for your rebuttal.
> >
> > I am still not convinced Gradient Coherent Hypothesis is a (sufficiently) novel contribution. For the sake of clarity, let us focus on (1) spectral bias hypothesis for why deep networks generalize, (2) Coherent Gradients Hypothesis being in some sense folk knowledge.
> >
> > First, spectral bias paper proposes a causal picture, rather than a descriptive picture, of why deep networks generalize. Formally speaking, the low frequency part of the function from x to y is, by the definition, *shared* between examples, and hence learned faster. That’s how Fourier transform works mechanistically speaking. Hence, Coherent Gradient Hypothesis seems to me to be very closely related to spectral (fourier) interpretation of why (which is a causal argument) deep networks generalize.
> >
> > To comment on the connection to linear regression. Sorry that I used work “mode”, which might have been not clear. By mode I refered to an eigenvector of XX^t in linear regression. By a very similar argument as in spectral bias perspective on generalization of DNNs, one can argue that the top eigenvector of XX^t is learned fastest in linear regression (i.e. error projected onto it reduces fastest) because it is shared between examples. See for example https://arxiv.org/pdf/1703.10622.pdf. Again this is almost the same as Coherent Gradients Hypothesis.
> >
> > Second, I still think there is an argument to be made that Coherent Gradients Hypothesis is folk knowledge. For instance, consider convolutional neural networks. It seems to me that one way to explain role of convolutional neural network (CNN) is that it biases the learning by making examples more similar to the learning procedure. For instance, if two examples have blue background, they will be more similar in terms of hidden activations to a convolutional neural network at initialization, than to a fully connected network (FC). This is very close to saying that gradients will be more similar for the two examples if CNN is used instead of FC, which is again very similar to Coherent Gradients Hypothesis.
> >
> > All in all, it seems to me that Coherent Gradient Hypothesis is too close to a rephrasing of already formulated hypotheses on why (causally speaking) deep networks generalize well, and I disagree that (all) the papers I mentioned are just descriptive.

---

> > > ### Author Response · Authors · 2019-11-11
> > > **Folk knowledge is unrebuttable :-)**
> > >
> > > Thank you for the quick response. Let's split into the two objections in two comments for your points  (1) and (2).
> > >
> > > For (2): Unfortunately, it is really hard to argue against a claim of folk knowledge for novelty since one can always claim any new work is folk knowledge. We can only note that no one we have shared this paper with (and many are experts in the field using deep learning practically on a daily basis or doing research) have said this is folk knowledge. Furthermore, folk knowledge does not come with experiments.
> > >
> > > We hope you appreciate the position you are putting us in. We humbly suggest that we focus only on published work and relating our work to that. (And through those references hopefully folk knowledge will come through.) But please let us know if this doesn't sound right.

---

> > > ### Author Response · Authors · 2019-11-11
> > > **Many differences with Spectral Bias paper**
> > >
> > > Thank you for the quick response. Let's split into the two objections in two comments for your points  (1) and (2).
> > >
> > > For (1), here are some concrete differences (we would appreciate a point by point response if you can spare the time):
> > >
> > > (a) Our explanation is much simpler (comes from properties of simple vector addition), and does not "exploit the piece-wise linear structure of ReLU networks" (a non-trivial assumption that allows them to use the heavy machinery of Diaz et al. 2016). Yet, in our reading, we do not see the simple fact about vector addition amplifying the common direction in their paper. But let us know if that is not right.
> > >
> > > This simplicity makes our approach more general, i.e., likely easier to extend to non-ReLU networks (sigmoids), to attention mechanisms, stochastic neurons, etc. and even general differentiable graphs.
> > >
> > > (b) They show empirical evidence of spectral bias. This is good, and we see that as an additional implication of Coherent Gradients. But as far as their experiments go, they are very different from the experiments we run.
> > >
> > > (c) Causality. As far as we can tell, they do not talk about *why* there is spectral bias (though would love pointers to specific sections/statements in the paper to that effect). Consequently they do not run experiments to "kill" the bias by modifying SGD. In contrast, we can run intervention experiments (Section 3) that do not appear in that paper (or anywhere else in the literature to our knowledge).
> > >
> > > Thus we are able to demonstrate that our approach is causal.
> > >
> > > Finally, on a more a more technical note, it is not immediately obvious to us if they can explain adversarial initialization (Liu et al 2019) whereas we believe our approach can, though we need to think about this more. A deeper question here is does the spectral bias in their analysis depend on the initialization? In our case it is clear that it does since the similarity metric is due to the initialization. Furthermore, it is not immediately obvious to see how their approach extends to situations where the inputs are very discrete/combinatorial whereas detecting common patterns still makes sense. [This is more evidence that the viewpoints are rather different.]
> > >
> > > We really appreciate you taking the time to press us on explaining how our paper is novel. We'll update the paper with these discussions. We hope our explanation gives you more confidence about the novelty. We really look forward to hearing from you.
> > >
> > > [PS: Somewhat off-topic, but what is your take on reconciling their approach (which doesn't seem to require stochasticity) and your belief (if we understand correctly) that stochasticity is necessary for generalization? If you have time to discuss, perhaps we should do that in a different thread.]

---

> > > > ### Comment · AnonReviewer3 · 2019-11-11
> > > > **Spectral bias paper puts forward a causal picture, and effectively (though not directly) seems to propose Coherent Gradient Hypothesis, and more on linear regression.**
> > > >
> > > > Thank you for the reply.
> > > >
> > > > Could you please also comment on my clarification of the relationship between linear regression, and Coherent Gradient Hypothesis?
> > > >
> > > > [To answer PS. I do not believe "stochasticity is necessary for generalization". The reason is that linear regression trained using GD "generalizes". Actually, the fact models trained using GD do not memorize fully data is trivially true for some models such as linear regression or random forests.]
> > > >
> > > > Most important, while I do agree that a large (key) part of the spectral bias paper is descriptive, the point is that they do propose causal explanation. Note that in Section 3 they propose/hypothesize (https://arxiv.org/pdf/1806.08734.pdf) that
> > > >
> > > > "However, Lf is bounded
> > > > by the parameter norm, which can only increase gradually
> > > > during training by gradient descent. This leads to the higher
> > > > frequencies being learned (...)", and
> > > >
> > > > "Second (cf. Appendix C.4), the exact form of the Fourier
> > > > spectrum yields that for a fixed direction kˆ, the spectral
> > > > decay rate of the parameter gradient (...)",
> > > >
> > > > where the second in C.4 a more rigorous argument. This more rigorous argument I think was made clearer in recent work such as https://arxiv.org/pdf/1901.06523.pdf. Therefore spectrum bias paper, and followup, do seem causal to me, and it seems to me they effectively propose a variant of Coherent Gradient Hypothesis.
> > > >
> > > > PS. Sure, I agree focusing on published work will be more productive

---

> > > > > ### Author Response · Authors · 2019-11-12
> > > > > **Please let us know if you agree with the differences with Spectral Bias**
> > > > >
> > > > > Thank you for the clarification.
> > > > >
> > > > > We think your comment on linear regression may be correct and we'd be happy to add to the discussion (assuming it holds upon further reflection).
> > > > >
> > > > > We would love your take on the detailed differences outlined in our original comment ("Many differences with Spectral Bias paper") w.r.t. novelty.
> > > > >
> > > > > And to reiterate our position: We agree that Spectral Bias is very relevant related work, and we will discuss it prominently based on this conversation, but we feel there are significant differences with it to merit acceptance of this paper. If you agree with the substance of this, please let us know if adding this detailed discussion is sufficient or not. Thanks again for your patience with this.

---

> > > > > > ### Comment · AnonReviewer3 · 2019-11-13
> > > > > > **Comments**
> > > > > >
> > > > > > Sure, let me answer a-c (the point about adversarial initialization seems a bit tangential, but let me know if you would like to discuss it)
> > > > > >
> > > > > > > (a) Our explanation is much simpler (comes from properties of simple vector addition)
> > > > > >
> > > > > > It is indeed simpler. However, my main point is that Gradient Coherence Hypothesis seems to be effectively proposed by them, but in a more formal language. It has the benefit that it enabled the authors to prove some Theorems/Propositions.
> > > > > >
> > > > > > > (...) (c) Causality. As far as we can tell, they do not talk about *why* there is spectral bias (though would love pointers to specific sections/statements in the paper to that effect)
> > > > > >
> > > > > > While I agree that you test empirically, they do talk about the "why" (see my previous answer). Arguably, both papers present rather weak empirical evidence; a simple experiment on MNIST might not be very predictive of larger models.
> > > > > >
> > > > > > To sum up, I still believe that the spectral bias paper effective proposes a causal mechanism for why deep networks generalize, which is very closely related to Gradient Coherence Hypothesis (see my previous answer)

---

> > > > > > > ### Author Response · Authors · 2019-11-13
> > > > > > > **Thanks**
> > > > > > >
> > > > > > > Thank you for this.
> > > > > > >
> > > > > > > We remain unconvinced that they effectively proposed Gradient Coherence Hypothesis since the simple insights from our paper (and the consequences of that) are not present in their work.
> > > > > > >
> > > > > > > The mathematical formalism in that paper is limited to computing the spectrum of a ReLU network but they do not prove any bounds (nice summary here: https://openreview.net/forum?id=r1gR2sC9FX). In contrast, we believe the connection to stability in our paper based on our simpler starting point is more promising as an approach to argue generalization.
> > > > > > >
> > > > > > > But of course, as mentioned before, we will put in a detailed discussion to the Spectral Bias work in our revised paper since we agree it is highly relevant. Assuming we make these revisions based on our discussion, do you still think our paper lacks novelty (given our discussion about the different starting point, simplicity/generality, connection to stability, completely different experiments, modification to SGD suggested by the theory) and should not be published?
> > > > > > >
> > > > > > > Thanks again for your time on this.

---

> > > > > > > > ### Comment · AnonReviewer3 · 2019-11-14
> > > > > > > > **Yes**
> > > > > > > >
> > > > > > > > I do believe the work is not correctly positioned in the literature, and in the current form should not be accepted. As far as I can tell, the proposition I cited from the Spectral Bias paper can be interpreted as putting larger weight on examples that share gradients. Arguably, the key idea behind the Spectral Bias paper, as well as of some prior work such as https://arxiv.org/abs/1706.05394, is that some examples are easier to learn and learn faster than random examples. And these papers are not just descriptive, but include causal arguments. Especially the Spectral Bias paper, as I argued in the discussion.
> > > > > > > >
> > > > > > > > At the same time, I would like to say that I like the paper overall. A resubmission that clarifies the contribution (as I argued, in my opinion as a simplification/improvement upon prior work), and then includes larger scale experiments to back up the value of the contribution, would definitely make for a strong paper.

---

### Official Review · AnonReviewer1 · 2019-10-22
**Official Blind Review #1**

**Rating:** 8

**Review:**

This paper posits that similar input examples will have similar gradients, leading to a gradient "coherence" phenomenon. A simple argument then suggests that the loss should decrease much more rapidly when gradients cohere than when they do not. This hypothesis and analysis is supported with clever experiments that confirm some of the predictions of this theory. Furthermore, since, as the authors emphasize, their hypothesis is prescriptive, they are able to suggest a novel regularization technique and show that it is effective in a simple setting.

I find the coherent gradient hypothesis to be simple and reasonable. Furthermore, the paper is written very clearly, and as far as I know the main idea is original (although since it is a rather simple phenomenon, it's possible something similar could have appeared elsewhere in the literature). Perhaps more importantly, the associated experiments are very cleverly designed and are very supportive of the hypothesis. For instance, Figure 1 provides compelling evidence for the coherent gradient hypothesis and in particular motivates the way phenomenon of early stopping arises a natural consequence. Overall, the paper is of very high quality, and I recommend its acceptance.

One criticism perhaps is whether these results are sufficiently significant. On the one hand, most of the experiments were done on small network and dataset combinations -- and the proposed regularization scheme as is will not scale to practical problems of interest. On the other hand, I really feel like I learned something interesting about gradient descent from reading this paper and absorbing the experimental results -- which is often not something I can say given the large array of reported experimental results in this field. It's clear that the authors themselves are aware that it's of interest to extend their results to more realistic settings, and regardless I think that this paper stands alone as is and should be accepted to ICLR.

**Experience Assessment:**

I have published one or two papers in this area.

**Review Assessment: Checking Correctness Of Derivations And Theory:**

I carefully checked the derivations and theory.

**Review Assessment: Checking Correctness Of Experiments:**

I assessed the sensibility of the experiments.

**Review Assessment: Thoroughness In Paper Reading:**

I read the paper at least twice and used my best judgement in assessing the paper.

---

> ### Author Response · Authors · 2019-11-10
> **Thank you**
>
> Thank you for your encouragement and we hope that our work inspires others to study this line of argument for understanding generalization. (We are also working on larger scale studies as mentioned in the Future Work and so far our experiments on other datasets have been encouraging.)

---

### Official Review · AnonReviewer2 · 2019-10-24
**Official Blind Review #2**

**Rating:** 8

**Review:**

Summary
The surprising generalization properties of neural networks trained with stochastic gradient descent are still poorly understood. The present work suggests that they can be explained at least partly by the fact that patterns shared across many data points will lead to gradients pointing in similar directions, thus reinforcing each other. Artefacts specific to small numbers of data points however will not have this property and thus have a substantially smaller impact on the learning. Numerical experiments on MNIST with label-noise indeed show that even though the neural network is able to perfectly fit even the flipped labels, the "pristine" labels are fittet much earlier during training. The authors also experiment with explicitly clipping "outlier gradients" and show that the resulting algorithm drastically reduces overfitting, thus further supporting the coherent gradient hypothesis.

Decision
The present work proposes a plausible, simple mechanism that might be contributing to the generalization of Neural Networks trained with gradient descent. Parts of the discussion stay informal as the authors themselves admit, but I appreciate that rather than providing mathematical decoration the authors focus on well-designed experiments that support their claims. Overall, the paper is of high quality and provides an interesting perspective on an important topic, which is why I think it should be accepted.

Questions for the authors
The coherent gradient hypothesis seems equally valid in the absence of stochasticity. However, the latter is often seen as an explanation of the generalization performance of SGD. My understanding is that you are also using minibatched gradient descent. Would you expect your experiments to still be valid when using deterministic gradient descent (full batch)? Did you study the effects of large batch sizes on the experiments?

**Experience Assessment:**

I do not know much about this area.

**Review Assessment: Checking Correctness Of Derivations And Theory:**

I assessed the sensibility of the derivations and theory.

**Review Assessment: Checking Correctness Of Experiments:**

I assessed the sensibility of the experiments.

**Review Assessment: Thoroughness In Paper Reading:**

I read the paper at least twice and used my best judgement in assessing the paper.

---

> ### Author Response · Authors · 2019-11-10
> **Thank you**
>
> Thank you for your feedback and the encouragement.
>
> Your question is very insightful. Indeed we believe that one of the predictions/consequences of Coherent Gradients is that stochasticity is not fundamental to generalization. Our experiments bear that out (we allude to it in Section 4 “from our experiments ..  it appears not to be necessary.”). These are some results with full batch training using otherwise the exact setup as Section 2 (thus each step is also an epoch). We see that in the 0% label noise case, there is good generalization throughout.
>
> Label Noise = 0%,     Step =     200, Training accuracy = 0.918, Test accuracy = 0.921
> Label Noise = 100%, Step =     200, Training accuracy = 0.125, Test accuracy = 0.110
>
> Label Noise = 0%,     Step = 10,000, Training accuracy = 0.996, Test accuracy = 0.981
> Label Noise = 100%, Step = 10,000, Training accuracy = 0.396, Test accuracy = 0.104
>
> Label Noise = 0%,     Step = 170,000, Training accuracy = 1.000, Test accuracy = 0.984
> Label Noise = 100%, Step = 170,000, Training accuracy = 1.000, Test accuracy = 0.108
>
> Note that this is also consistent with the findings in the recent large scale study (https://arxiv.org/abs/1811.03600) who find “no evidence that larger batch sizes degrade out-of-sample performance.”

---

> > ### Comment · AnonReviewer2 · 2019-11-14
> > **Thanks for the additional experiments**
> >
> > Thank you for the additional clarification and experiments.

---

### Public Comment · ~Stanislav_Fort1 · 2019-10-23
**A paper with a very significant overlap from January 2019 at arxiv.org/abs/1901.09491**

I enjoyed reading your paper. I think that observing the correlation between gradients on different inputs / minibatches is potentially a very fruitful direction to explore.

I am one of the co-authors of "Stiffness: A New Perspective on Generalization in Neural Networks" (https://arxiv.org/abs/1901.09491 ) which we uploaded to arXiv in January 2019 (9 months ago). It appears that our paper has a very significant overlap with the content of yours.

In addition, several papers citing our submission and building up on the ideas in it (to some extent) appeared prior to your OpenReview submission. For example in arxiv.org/abs/1904.06963, a very similar metric to yours is used to quantify what they call the "gradient confusion".

It would be great if you had a look at our paper. Provided you found it related, we would appreciate if you cited our prior work on the topic.

---

> ### Author Response · Authors · 2019-10-28
> **Though both papers look at per-example gradients and generalization, it appears there are significant differences**
>
> Thank you for your comment. It is exciting to see that many of us look at per-example gradients. The time is ripe to explore this approach! A few questions before diving into a more technical discussion:
>
> (a) Please clarify, from your perspective, what is the overlap between your work and our work.  For us, it appears that there is no overlap other than the fact both focus on per-example gradients in some form or another.
>
> (b) Please clarify the relationship between the second paper (arxiv.org/abs/1904.06963) and your work.  For us, it appears that the second paper is an independent contribution, which references your work in the last sentence as a possible direction for future work.
>
> (c) Is our understanding correct that all three papers (yours, theirs, and ours) are concurrent submissions to this conference (https://openreview.net/forum?id=H1e31AEYwB and https://openreview.net/forum?id=r1xNJ0NYDH)? If so, let us hope all of them get in and we will have a great dedicated session :)
>
> That said, we enjoyed reading both papers, and we are very happy to cite your technical report and all relevant work. The more the merrier.
>
> Now, on a more technical note, we judge the primary difference between your work and ours as follows. You identify a *descriptive* metric (stiffness) and show that it correlates well with generalization. In contrast, we do NOT propose a metric. Our focus is on trying to understand *why* sometimes gradient descent generalizes and why sometimes it doesn’t. By analogy to random forest and decision tree construction, we try to find the mechanism by which SGD detects commonality -- the Coherent Gradients mechanism. Since our explanation is a causal one, we are able to go beyond descriptive statistics and provide an intervention experiment (Section 3) where using insights from Coherent Gradients (and using an analogy with Random Forests), we can naturally modify SGD to reduce overfitting. Furthermore, in line with our primary objective of trying to understand why generalization happens, we draw a connection to stability and advocate using data-dependent stability as the lens through which we should try to understand generalization.
>
> For us, the metrics are secondary in the sense that they are a means to building confidence in the Coherent Gradients hypothesis. We’d happily use stiffness (your proposed metric) or gradient confusion (from 1904.06963) to build confidence in the same, and it is interesting to consider to what extent your results build or deny further confidence in Coherent Gradients. Furthermore, even for our descriptive statistics, the methods and the statistics themselves (e.g. studying label noise (a method applicable to regression as well), the exact quantities that we look at (dot products of pristine and corrupt with *overall* gradients, their integral over time, their statistical significance; the first time an example is learnt, etc.) are quite different from what you have in your paper, and so may be viewed as complementary. If this doesn’t sound right, please do let us know precisely what you think the overlap is.
>
> Finally, the focus of arxiv.org/abs/1904.06963 is on optimization in contrast to generalization (which is the focus of both of our papers). The gradient confusion metric they propose to explain the speed of learning is much more similar to your metrics (relating to expectations of pairwise per-example gradients) than any we use (dot products with the overall gradient) and so we are confused by your statement that that is “a very similar metric to yours”, and would love for you to clarify what you mean. However, based on Coherent Gradients, we believe that there is a close connection between the speed of learning and generalizability which we briefly touch upon in Section 5 on wide networks.
>
> If any of this doesn’t sound right or fair, please let us know. Thanks.

---

### Author Response · Authors · 2019-11-15
**New version uploaded with updated discussion of related work**

We have uploaded a new version of the paper with detailed discussion of the papers brought to our attention since the initial submission. The bulk of the changes are in Section 4 ("Discussion and Related Work").

(There are some minor changes in Section 5 ("Future Work") and formatting adjustments elsewhere to stay within page limit given the added discussion.)

---

### Decision · Program_Chairs · 2019-12-19

**Decision:**

Accept (Poster)

**Comment:**

The paper proposes an intuitive causal explanation for the generalization properties of GD methods. The reviewers appreciated the insights, with one reviewer claiming that there was significant overlap with existing work.

I ultimately decided to accept this paper as I believe intuitive explanations are critical to the propagation of ideas. That being said, there is a tendency in this community to erase past, especially theoretical, work, for that very reason that theoretical work is less popular.

Hence, I want to make it clear that the acceptance of this paper is based on the premise that the authors will incorporate all of reviewer 3's comments and give enough credit to all relevant work (namely, all the papers cited by the reviewer) with a proper discussion on the link between these.